# Effect of Controlled Microbial Fermentation on Nutritional and Functional Characteristics of Cowpea Bean Flours

**DOI:** 10.3390/foods8110530

**Published:** 2019-10-25

**Authors:** Luis M. M. Ferreira, Ana Mendes-Ferreira, Clícia M. J. Benevides, Diana Melo, Anabela S. G. Costa, Arlete Mendes-Faia, Maria Beatriz P. P. Oliveira

**Affiliations:** 1CITAB-Centre for the Research and Technology of Agro-Environmental and Biological Sciences, 5001-801 Vila Real, Portugal; lmf@utad.pt; 2University of Trás-os-Montes and Alto Douro, 5001-801 Vila Real, Portugal; anamf@utad.pt; 3WM&B—Wine Microbiology & Biotechnology Lab, Department of Biology and Environment, University of Trás-os-Montes and Alto Douro, 5001-801 Vila Real, Portugal; 4Biosystems & Integrative Sciences Institute, Faculty of Sciences University of Lisbon, 1749-016 Lisboa, Campo Grande, 1749-016 Lisboa, Portugal; 5University of Estado da Bahia, Department of Life Sciences, Cabula, CEP 41150-000, Salvador-BA, Brazil; cbenevides@uneb.br; 6REQUIMTE, LAQV/Faculty of Pharmacy from University of Porto, Rua de Jorge Viterbo Ferreira, n.º 228, 4050-313 Porto, Portugal; melo_dian@hotmail.com (D.M.); anabelac020@gmail.com (A.S.G.C.); beatoliv@ff.up.pt (M.B.P.P.O.)

**Keywords:** cowpea beans flours, fermentation, single and mixed-culture, nutritional value, antioxidant activity

## Abstract

This study aimed to optimize bean flours fermentation through the use of appropriate technological procedure and, thereby, to obtain a high quality and safe product. In this line, cowpea bean flours with different moisture conditions (10, 20 and 30%) were incubated with (1) a single culture of *Lactobacillus plantarum*, or (2) a consortium of lactic acid and acetic acid bacteria, together with one strain of *Saccharomyces cerevisiae*. Effects of inoculation of cowpea beans flours on stability (i.e., evaluated by the decrease in pH), and variations in nutritional characteristics (i.e., protein, starch, water soluble carbohydrates, total dietary fibre) were investigated. In both fermented flours, the effect of fermentation was more noticeable in total water-soluble carbohydrate (WSC) concentration during the fermentation process (*P* < 0.001), accounted for by metabolic activity of the microorganisms. The pH values progressively decreased (*P* < 0.001) through the fermentation process, particularly in flours fermented with a single culture of *L. plantarum*. By contrast, titratable acidity increased (*P* < 0.001) throughout the fermentation process in F2 and F3, although more noticeable in F3. Total dietary fibre (TDF) was not variable over the time. In relation to the protein content, the fermentations behaved very similarly; F2 had a variation over the time, but the effect was not significant (*P* = 0.0690). Results revealed small changes in chemical composition except in the case of pH and sugar contents with the values lower in the fermented products (i.e., single- or mixed-culture fermentation), leading to a more stable and safety product. These results indicate that fermented dry beans flours have the potential as functional ingredients for new food formulations.

## 1. Introduction

The importance of legumes in the world’s diet led the United Nations to declare 2016 as the international year of leguminous (grain legumes) as “nutritive seeds for a sustainable future” [1]. Cowpea (*Vigna unguiculata* L. Walp) belongs to the family *Fabaceae* originating from Africa and widely distributed throughout semi-arid sub-Saharan Africa, in rural areas of West and Central Africa, East Asia and South America and dry zones of Central America [2,3]. Global production of cowpea beans has remarkably increased worldwide in recent years, with the bean cultivated in more than 100 countries between 40° N and 30° S latitudes [4]. Cowpea ability to growth in low fertility and to subsist in soils where drought is a major constraint due to low and irregular rainfall [5,6,7], confers advantages over other legume crops. Additionally, it is an inexpensive source of protein (250–295 g·kg^−1^ DM), carbohydrates (694–859 g·kg^−1^ DM) [8,9], minerals and vitamins [10,11], essential nutrients for a balanced human diet. Its high nutritive value, associated to low fat and high protein and fibre, suggests its potential application in weight restriction diets [1,12], and prevention of cardiovascular diseases [13]. 

For centuries, food preservation by fermentation has been practiced in several raw materials including dairy and non-dairy products, particularly in the former [14]. Different groups of microorganisms are involved in the process, mainly food-grade bacteria, yeast and, to a certain extent, moulds [14]. Fermentation process confers microbiological stability and increases the nutritional value and/or functional characteristics [15] and modifies its organoleptic characteristics. Currently, fermented foods unveil a beneficial effect in altering the gut microbiome through a probiotic effect. By contrast, fermentation of non-dairy matrices, the utilization of starters and type of inoculum, survival of microorganisms and how they cope with stressful conditions, or their relationship with natural microorganisms still need to be elucidated. Few studies suggest that microbial fermentation in grain legumes substantially improves the nutritional value of legumes, by increasing protein digestibility [16], palatability and the levels of B group vitamins [17] and significantly reduce anti-nutritional factors as phytates, trypsin inhibitor activity, saponins, tannins and raffinose oligosaccharides [16,17], flatulence-causing oligosaccharides α-galactosides [18,19]. It is well known that wet conditions in a food often induce microbial growth and contribute in chemical and enzymatic reactions and spoilage process [20,21]. The amount of free water, called water activity, affects food stability, chemically and microbiologically [20,21]. 

The main goal of this study was to obtain a new product without decreasing the nutritional value of the raw-material and, simultaneously, introducing new characteristics resulting in a more stable and safe product, by a very well controlled microbial fermentation. We anticipate that the results obtained in this study can contribute for an additional use of this legume turned into an added-value product to the family economy.

## 2. Materials and Methods 

### 2.1. Reagents and Standards

Gallic acid, tannic acid, epicatechin, trolox, sodium acetate, Folin–Ciocalteu’s phenol reagent, DPPH• (2, 2-diphenyl-1-picrylhydrazyl), sodium nitrite, ferric chloride, aluminium chloride, TPTZ (2,4,6-tripyridyl-s-triazine) solution, and ferrous sulfate heptahydrate were all obtained from Sigma–Aldrich (St. Louis, MO, USA). Sodium carbonate anhydrous, sodium hydroxide and absolute ethanol were purchased from Merck (Darmstadt, Germany). Ultrapure water was treated in a Milli-Q water purification system (Millipore, Bedford, MA, USA) and used to prepare all aqueous solutions.

### 2.2. Preparation of Cowpea Bean Flours

Cowpea beans were purchased from a local supermarket and were sorted, cleaned of impurities and selected the largest and entire beans and then ground to pass 2 mm sieve (Retsch, cutting mill, model SM1, 42781 Haan, Germany), to obtain a granular product with small particle size and greater surface area that microbiota can interact with and thus increase its fermentability. Ground beans (70 g) were dispensed into small screw caps glass-flasks which were autoclaved, for 10 min at 65 °C or 100 °C (60L- stainless steel steam autoclave, Orion B6, GA, USA). Major steps of flours fermentation process are displayed in Figure 1. Immediately after the thermal shock, the flasks were cooled on ice (about 15 min) and 125 μL of 6% sulphur dioxide solution/100 g of seeds was added to each flask under aseptic conditions. Three levels of moisture (10, 20 and 30%) were investigated in the four set of biological independent experiments. In the first set, ground beans were soaked under aseptic conditions with sterile saline solution to obtain 10% of humidity and fermentation was conducted during 72 h. In the next set of experiments, fermentation length was extended to 96 h and moisture content increased to 20% with sterile saline solution. In the 3rd and 4th set of experiments, ground beans were soaked with saline solution to obtain 30% of humidity and fermentation length was maintained for 96 h. The 3rd set of experiments differs from the 4th because thermal shocks were conducted at different temperatures, at 65 or 100 °C for 10 min, respectively. Immediately after the thermal shock, the flasks were cooled on ice (about 15 min) and 125 μL of 6% sulphur dioxide solution/100 g of seeds was added to each flask under aseptic conditions. The same inoculation procedure was used in all experiments.

### 2.3. Microorganisms

As fermentation-starters cultures two strains of lactic acid bacteria (LAB) species, *Lactobacillus plantarum* (Lp), a facultative homofermentative LAB which converts glucose almost exclusively into lactic acid, and *Oenococcus oeni* (Lo), a heterofermentative LAB which catabolize glucose into lactic acid, ethanol/acetate and CO_2_ as well as yeast strain of *Saccharomyces cerevisiae* (Sc), that converts the sugar into ethanol CO_2_, and a strain of acetic acid bacteria, *Acetobacter aceti* (Aac), able to oxidise ethanol into acetate, all belonging to the microbial collection of the Wine Microbiology and Biotechnology Lab, at the University of Trás-os-Montes and Alto Douro (UTAD), were used. Bacteria have been stored in cryovials in a freezer at −80 °C in glycerol (30%) (ULT Freezer, Sanyo, MDF-U3386 S, Osaka, Japan). LAB were revitalized by inoculating into MRS (Man, Rogosa and Sharpe) broth and Aac was revitalized in GYC broth (Glucose 5%, Yeast Extract 1%, Calcium carbonate 2%, pH 6.8). During the experimental work, LAB were propagated into MRS broth and Aac were transferred into Brain Heart Infusion broth (BHI). The strain of *S. cerevisiae* used herein was maintained in YPD (Yeast extract 10 g, Peptone 20 g, Dextrose 20 g. Agar 20 g, for plates only). 

### 2.4. Inoculum Preparation and Fermentation Experiments

For all experiments, a starter culture was prepared by pre-growing each microorganism separately as follows: the yeasts in YPD broth, LAB in MRS broth and Aac in BHI broth. Pre-culture was used to inoculate the milled beans with an initial cellular concentration of 10^6^ mL^−1^. The fermentations were conducted in small screw caps glass-flasks filled to 2/3 of their volume. F1 (Control): The autoclaved milled beans were watered with saline solution (0.85% of NaCl) to keep similar moisture in each assay (10, 20 or 30%); F2 (Lp): After cooling in ice, the autoclaved milled beans were inoculated with *Lactobacillus plantarum* previously grown in MRS and cells were harvested by centrifugation (5000 g, 10 min, at 4 °C) and re-suspended in saline solution prior to inoculation. Cell suspensions were spread throughout the beans to keep similar humidity in each assay; F3 (Sc-Lp-Lo-Aac): In this formulation, a consortium of the same strain of LAB used in F2 experiment, *L.plantarum* (Lp) plus *O. oeni* (Lo), *S. cerevisiae* (Sc) and *A. aceti* (Aac) was used as inoculum. Each one was separately grown in appropriate culture media, MRS, YPD or BHI broth and prior to inoculation each one was centrifuged at 5000 g, for 10 min, at 4 °C (Sigma 3-18K refrigerated Centrifuge, 37520 Osterode am Harz, Germany). Supernatants were removed and cells were re-suspended in saline solution as described above. Similar moisture was maintained in each assay. After inoculation, all flasks were mixed by rotation and inversion, and incubated at 28 °C, for 24, 48 or 96 h. (Twice a day the flasks were shaken). Samples were collected daily and freeze for later analysis. All experiments were done in duplicate and repeated as independent biological replicates. 

### 2.5. Microbiological Analysis 

Microbiological analysis was carried out using the standard methodologies described for food analysis including, for the preparation, suspension and dilution of samples, enumeration of mesophilic lactic acid bacteria (LAB) ISO 15214:1998 [22] and yeasts/moulds ISO 21527-1:2008 [23]. Briefly, cowpea beans (10 g) were aseptically mixed with 90 mL of buffered peptone water (BPW) in a Stomacher bag and crushed in a stomacher (STAR Blender LB 400, VWR, Radnor, PA, USA) for 5 min to obtain a uniform suspension of the pulp material. One ml of the suspension was serially diluted in 0.1% peptone water and 0.1 ml or 1 mL samples from each of three consecutive dilutions were spread inoculated onto duplicate plates of different agar media. The enumeration of LAB was done by spreading 1 mL on a double-layered plate of de Man Rogosa Sharpe (MRS) agar (Liofilchem, Roseto degli Abruzzi, Italy) and incubated at 30 °C for 72 h. Yeasts/moulds were enumerated by spreading 0.1 mL of each dilution on agar plates of Dichloran Rose Bengal Chloramphenicol agar (DRBC) (Liofilchem) and incubated. After incubation at 25 °C for 72 h, the typical colonies of yeast or bacteria were counted, and data was expressed as log colony-forming units (CFU) g^−1^ sample. All analyses were done in triplicate.

### 2.6. Analytical Procedures

The samples collected at the end time point, after 24, 48, or 96 h were dried in a forced air oven at 60 °C, milled to pass through a 1 mm screen (Retsch SM1 Cutting Mill, Haan, Germany) and stored in air tight flasks at room temperature for subsequent chemical analysis. Samples were analysed for dry matter, organic matter, ash, protein and crude fat content, according to the methods of the Association of Official Analytical Chemists [24]. Total N was assessed as Kjeldahl N (nº 954.01, AOAC, 2006) using a Kjeltec System 1026 Distillation Unit (Velp, Usmate, Italy). Protein was calculated as Kjeldahl N × 6.25. The total water-soluble carbohydrate (WSC) was determined by the anthrone method [25]. Briefly, soluble sugars were extracted with 800 mL·L^−1^ ethanol from 100 mg of sample in a water bath. Next, 3 mL of anthrone solution was added to 200 μL of sample extract and heated in a water bath at 100 °C. Standard curves were prepared with stock glucose solutions. Finally, the absorbance of solutions at 625 nm was read in a spectrophotometer (Shimadzu UV mini 1240, Kyoto, Japan). Crude fat content was determined by extracting the fat using an organic solvent (petroleum ether) in a Soxhlet apparatus (JP Selecta, 08630 Abrera, Barcelona, Spain). Total dietary fibre, including soluble and insoluble fractions, were determined using an enzymatic assay procedure (K-TDFR 04/17, Megazyme, Bray, Ireland) following the procedures of AOAC (2006, nº 991.43) [24]. All analyses were performed in triplicate and results are expressed as g per 100 g.

The titratable acidity, expressed as g·L^−1^ of lactic acid, was determined by the standard AOAC method [24]. Briefly, 10 g of fermented flours were suspended in 75 mL of distilled water and allowed to macerate for 30 min. The mixture was filtered, and 10 mL aliquots were titrated with 0.1 N NaOH using a phenolphthalein indicator for end-point determination. The suspension mixture was used for measurement changes in the pH using a Crison 2002 pH meter and lactic acid. Lactic acid was quantified using an enzymatic assay procedure (K-DLATE 07/14, Megazyme, Bray, Ireland). 

### 2.7. Determination of Total Phenolic Content, Total Flavonoid Content and Antioxidant Activity 

#### 2.7.1. Hydro-Alcoholic Extracts Preparation

The solvent extraction was performed by maceration with 1 g of the powdered samples with 20 mL of 1:1 ethanol/water at 40 °C for 30 min (Mirac, Thermolyne, Dubuque, IA, USA). The extracts were filtered through Whatman no. 1 filter paper, then concentrated under vacuum at 37 °C until dryness and stored at 4 °C until further use. The obtained extracts were used for determination of total phenolic content, total flavonoid content and antioxidant activity.

#### 2.7.2. Determination of Total Phenolic Content

Total phenolic content (TPC) was spectrophotometrically determined following the Folin-Ciocalteu procedure [26] with minor modifications [27]. Briefly, 30 μL of extract was mixed with 150 μL of Folin–Ciocalteu reagent (10× dilution) and reacted for 5 min. Then, 120 μL of Na_2_CO_3_ 7.5% solution was added and kept at 45 °C for 15 min, and at room temperature for 30 min. The absorbance was read at 765 nm in a Synergy HT Microplate Reader (BioTek Instruments, Inc., Winooski, VT, USA). Gallic acid (5–100 mg·L^−1^) was used as standard to prepare the calibration curve in order to obtain a correlation between sample absorbance and standard concentration. TPC was expressed as µg of gallic acid equivalents (GAE) per 100 mg of flour dry matter (DM).

#### 2.7.3. Determination of Total Flavonoid Content

Total flavonoid content (TFC) was colorimetrically determined based on flavonoid °SL3 0A Taluminum compounds formation [28]. Briefly, 1 mL of extract was mixed with 4 mL of water and 300 μL of 5% (*w*/*v*) NaNO_2_ solution. After 5 min, 300 μL of 10% (*w*/*v*) AlCl_3_ solution was added, and after 1 min, 2 mL of 1 mol·L^−1^ NaOH and 2.4 mL of water were added as well. Afterwards, the absorbance was read at 510 nm in a Synergy HT Microplate Reader. Catechin was used as standard to obtain the calibration curve (0–400 μg·mL^−1^). TFC was expressed as µg of catechin equivalents (CAE) per 100 mg of flour DM.

#### 2.7.4. Determination of Antioxidant Activity 

DPPH• (2,2-diphenyl-1-picrylhydrazyl) Free Radical Scavenging Assay

In order to determine the DPPH• free radical scavenging assay of the extracts, the method described by Costa et al. [28] was followed. Briefly, 30 µL of sample, 270 µL ethanol solution, containing DPPH• radicals (6.0 × 10^−5^ mol·L^−1^ in ethanol). The decrease of the DPPH• radical was measured every 2 min by monitoring the decrease of absorption at 525 nm (microplate Synergy HT Reader; BioTek Instruments, Synergy HT GENS5, EUA), allowing to obtain the kinetics reaction. The reaction endpoint was achieved in 20 min. A calibration curve was prepared with trolox (5.62–175.34 mg·L^−1^). The results were expressed as µg of trolox equivalents (TE) per 100 mg of flour DM.

Ferric Reducing Antioxidant Power (FRAP) Assay

A FRAP assay was determined following Benzie and Strain [29] method with minor modifications by Costa et al. [28], which is based on the reduction of a ferric 2,4,6-tripyridyl-s-triazine complex (Fe^3+^-TPTZ) to the ferrous form (Fe^2+^-TPTZ). Briefly, 35 μL of the extract were added to 265 μL of FRAP reagent (10 parts of 300 mM sodium acetate buffer at pH 3.6, 1 part of 10 mM TPTZ solution and 1 part of 20 mM FeCl_3_·6H_2_O solution), the reaction mixture was incubated at 37 °C. After 30 min, the increase in absorbance was measured at 592 nm. A calibration curve was prepared with ferrous sulphate (FeSO_4_·7H_2_O, linearity range: 150–2000 µM). The results were expressed as µmol of ferrous sulphate equivalents (FSE) per 100 mg of flour DM.

### 2.8. Statistical Analysis

Variation of the chemical composition of the fermented cowpea flours was examined using a MANOVA procedure for repeated-measures (JMP, SAS Institute, Cary, NC, USA), including the effects of fermentation type (FT). Tukey’s test was used for multiple mean comparisons at a significance level of 0.05. Prior to the analysis, normality of the data was checked using Shapiro–Wilk test. Only TA, protein and starch data did not follow the normal distribution and, consequently, they were log_10_-transformed before the MANOVA procedure.

## 3. Results and Discussion

### 3.1. Chemical Composition and Nutritional Value of Cowpea-Beans

Comparative analysis of nutritional composition of cowpea beans flours and others from similar legumes are presented in Table 1. Data on the nutritional characteristics of unfermented cowpea-beans flours obtained in this study is consistent with that reported in previous studies for this and other types of legume [9,17,30]. However, the levels of moisture were quite low compared to the values obtained in legumes from other origins, which may explain the slow decrease of pH observed in this study.

Mean values of protein content of unfermented cowpea-bean flours (254.2 ± 8.3 g·kg^−1^ DM) are in accordance with most of the values reported from others [9,10,17,31,32], with the exception of the protein content in *Vigna unguiculata* L. Walp [11] and in chickpea and cowpea beans [31]. According to Kachare et al. [33] and Sosulski et al. [34], 5–37.0% of the total protein in cowpea (mainly globulins) are nutritionally unavailable [33,34]. Thus, one can expect that enzymatic proteolysis carried out by microbial activity can be potential valuable as functional food ingredient. 

Values of total fat content (17.7 ± 6.1 g·kg^−1^ DM, Table 1) were comparatively lower compared to those reported for other legumes (chickpea, split pea, lentil, green gram and lupine), ranging from 21 to 79.8 g·kg^−1^ DM, respectively in *Vigna unguiculata* L. Walp, [11] and in a local species of cowpea (*Vigna biflorus* and *Vigna sinensis*; local names: oraludi and apama) [10]. 

Total carbohydrate content found in this study (418 g·kg^−1^ DM) is lower than that reported in the literature, averaging 493 g·kg^−1^ DM (Table 1). These discrepancies can be explained by differences on the methodology used, most of studies determined this parameter by difference. In this study, total starch and water-soluble sugars were measured separately. Granito et al. [17] found even lower values, ranging from 243 to 297 g·kg^−1^ DM of total starch, depending on the variety, being about 50% and 68% correspondent to resistant starch, significantly higher in cowpea flour of Tuy variety.

Mean values of TDF content found in this study are 133.6 ± 31.7 g·kg^−1^ DM, values in accordance to those reported in the literature (141–194 g·kg^−1^ DM) [17,31]. 

Dietary fibre consists of non-digestible carbohydrates and lignin that are intrinsic and intact in plants, being recommendable its daily base intake in adults and children [35]. Thus, the values of TDF linked to the low values of fat content of cowpea beans and potential levels of resistant starch suggest a potential benefit in calorie restriction diets. In general, legumes are an interesting source of dietary fibre and also resistant starch; their content showed an inverse correlation with predicted glycaemic index [36].

### 3.2. Effects of Fermentation on Stabilization of Cowpea Bean Flours

Growth of microorganisms in food is dependent on intrinsic factors such as nutrient content, water activity, pH value, redox potential and extrinsic factors such as the temperature of storage. It is well known that moisture conditions in a food often induce microbial growth, and contribute to chemical and enzymatic reactions and spoilage processes. The amount of free water, called water activity, affects food stability, chemically and microbiologically. Also, acidification of food by addition of organic acids or by microbial fermentation along with low water activity result in products that are more stable and less susceptible to microbial spoilage. Thus, the effects of fermentation, the type of inoculum and the level of moisture on the content of nutrients (i.e., protein, starch, water soluble carbohydrates, total dietary fibre, fat, among others) and on microbial stability of cowpea bean flours were investigated.

Through the experimental work, three levels of moisture in the beans (i.e., 10%, 20% or 30%) were also tested. At the end of the fermentation process, the microbiological control of fermented and non-fermented flours, was performed. 

Preliminary experiments conducted with 10% of moisture led to levels of pH within 6.30–6.44 in the inoculated and non-inoculated flours, respectively, suggesting that under this humidity level the microorganisms were unable to ferment available sugars (Ferreira and Mendes-Ferreira, unpublished results). In the next set of experiments, fermentation length was extended and moisture content increased to 20%. The results are presented in Table 2. Under this humidity level, pH values ranged from 5.26–5.02 to 5.97, after 72 h, respectively, in the inoculated and non-inoculated flours, accompanied by a slight decrease in sugar content in F2. Results suggest that 20% of humidity still constrains the growth and microbial performance.

Thus, new trials were conducted in which the fermentation length was extended to 96 h and moisture content increased to 30%. The results are presented in Table 3 and indicate that, under this moisture level, a stronger acidification of the flours was obtained, pH values decreased from 6.0 to 4.65 in flours inoculated with the single-culture of *L. plantarum* after 4 days of fermentation. This decrease was accompanied by a more noticeable decrease in the sugar content in both F2 and F3 assays, compared to the control. The drop in pH resulted in an increase of total acidity of the fermented flours. The values of pH obtained here were quite different from those reported by Frías et al. [18] in which the decrease in pH (from 6.9 to 3.9), was observed in the first 24 h of natural fermentation of lentils. These differences can be due to several factors: 1) fermentation of lentils was conducted with epiphytic microorganisms present on the seeds at higher temperature (28 °C, 35 °C, 42 °C), and 2) higher water activity (concentration of 100 g of lentil flour/L of water)and 3) also by the strains used as fermentation starters in this study.

Based on the results, new trials were conducted in which ground beans were dispensed into small screw caps glass-flasks were autoclaved, for 10 min at 100 °C, before the inoculation, using the same procedure already described. Results are presented in Table 4. Under these conditions, no LAB growth was observed on MRS agar from the non-inoculated assay.

### 3.3. Effects of Fermentation with Single or Mixed-Culture on Chemical Composition of Cowpea-Beans Flours

It is generally accepted that microbial fermentation increases the nutritional value of legumes, by increasing the levels of essential nutrients and/or by reducing the level of anti-nutrients in food [17,18,37]. Most of studies on legume fermentation have been conducted using the natural microbiota i.e., epiphytic microorganisms present in seeds [18,36], in different varieties of *Vigna sinensis* and subsequent cooking process [17]. 

To our knowledge, this is the first study to describe the effects of very well controlled fermentation, conducted by single- or mixed-cultures, on the nutritional value and stability of cowpea beans flours.

In Table 3 and Table 4 the nutritional characteristics of cowpea-beans flours (DM, OM, Protein, starch, soluble sugars, among others) fermented under conditions of 30% of moisture are presented. 

The concentration of starch in flours varied (*P* = 0.0074, Table 5) depending on the type of inoculum used: in F3 and F1 (control) the concentration of starch slightly increased, while in the F2 assay (inoculated with a single culture of *L. plantarum*), it decreased. The interaction between the two variables was not significant (*P* = 0.0915). Similar findings have been reported by Granito et al. [37] when submitting *Phaseolus vulgaris* to natural fermentation after 48 h at 42 °C. Starch may serve as a substrate for acids production by epiphytic microorganisms during the early stages of fermentation.

The decrease in WSC concentration during the fermentation process was clear (*P* < 0.001). Although the differences between experiments, at all fermentation times the WSC content were higher in the controls. In fact, there was a decrease in the WSC value over time in all fermentations, (less marked in the control) where microbial growth was undetectable. Indeed, sugar concentration decreased in both fermented assays F2 and F3, which is the result of the production of various organic acids such as lactic, the major acid produced in the F2 and lactic and acetic acids and ethanol in the course of fermentation in F3, due to the metabolic activity of the microbial consortium. The pH values progressively and significantly decreased throughout the fermentation process (*P* < 0.001). The deeper decrease was in the flours fermented with a single culture of *L. plantarum* (F2 experiment) followed by F3, inoculated with the microbial consortium. In F1, as expected, the variation in pH was negligible. Therefore, the interaction between the factors was also significant (*P* = 0.0115). In accordance with this, titratable acidity increased significantly (*P* < 0.001) throughout the fermentation process, in F2 and F3, although it was more noticeable in F3. 

There was a tendency for the decrease in DM values during fermentation, however, this trend was not significant (*P* = 0.5504). Significant differences were found in MS between experiments (*P* < 0.001) since the MS values were higher in the 2nd experiment (Table 4).

TDF contents have not changed during fermentation in all experiments (*P* = 0.8175) although differences between experiments, (*P* = 0.0492), have been detected since its values were higher in the first experiment (Table 3).

In relation to protein content variation, fermented flours were very similarly. The F2 fermentation had a variation over time, but the effect was not significant (*P* = 0.0690). A gradual increase was observed in the 1st experiment (Table 3), which was difficult to explain, when there was no significant variation in protein content in the 2nd experiment (Table 4). In this regard, results revealed small changes on chemical composition except on pH and sugars content which values were lower in the fermented products, in single- or mixed-culture fermentation, leading to a more stable and safety product.

### 3.4. Effect of Fermentation on Total Phenolic and Total Flavonoid contents and Antioxidant Activity of Cowpea Beans Flours 

The total phenolic and flavonoid contents and antioxidant activity of cowpea beans flours extracts were evaluated; the results are presented in Table 6 and Table 7. Two different assays were used to screen the antioxidant properties: reducing power, measuring the conversion of a Fe^3+^/ferricyanide complex to the ferrous form (Fe^2+^), and scavenging activity, measuring the decrease in DPPH• radical absorption after exposure to radical scavengers.

The TPC and TFC of the hydro–alcoholic extracts of the bean flours varied significantly (*P* < 0.05) among the treatments. The highest values of TPC and TFC were found in the hydroalcoholic extracts of the bean flours inoculated with *L. plantarum* (F2) which showed high extraction yield in the first set of experiments (Table 6), while the second set was in accordance with the values shown in non-inoculated flours (control). The antioxidant activities of the F2 flours also showed the highest antioxidant activity using both the DPPH• inhibition and FRAP assay, which is in agreement with the values obtained for TPC and TFC in the same bean flours. Conversely, in the second set of experiments (Table 7), the highest values of TPC and antioxidant activity measured by DPPH• scavenging assay were observed in the flours fermented with the consortium of microorganisms (F3). 

Comparing the values of TPC and TFC of non-inoculated flours with the results obtained for the cowpea bean grain, values of TPC were approximately six-fold higher, while the values of TFC were about five-fold lower than those reported [38]. It should also be pinpointed that the values of the in vitro antioxidant activity of the DPPH• scavenging assay found in this study are in accordance with those reported for cowpea beans by Zhao et al. [38]. 

Moreover, the decrease in TFC in F3 over time could also decrease the amount of tannins in the cowpea beans flour, which are considered as antinutritional factors, due to their astringency and therefore, the flavour of the flour could be enhanced. It may also have decreased other antinutrients, such as trypsin inhibitors, raffinose, stachyose and phytic acid [39].

## 4. Conclusions

The inoculation of cowpea bean flours with a single culture or a consortium of microorganisms revealed how fermentation influenced the physicochemical and possibly the functional properties of this new product. In fact, in this study, pH, total acidity, dry matter, TDF, WSC and protein were found to be altered in cowpea bean flours. Also, phytochemicals content, namely, phenolic compounds and flavonoids, and the results for antioxidant activity assays suffered changes with microbial fermentation. The high content of dietary fibre of this product can have numerous therapeutic and beneficial effects in the organism due to stimulation of the gastrointestinal microbiota, functioning as a prebiotic and enhancing human health. Moreover, it is a plant-based protein source which meets current trends in human diets with increasing interest in natural products incorporation and following flexitarianism, vegetarianism or veganism food patterns. It may therefore be nutritionally interesting to ferment cowpea beans before their use as a functional ingredient, since the fermentation process produced a more stable and safer product with the potential to be incorporated in new foodstuffs.

## Figures and Tables

**Figure 1 foods-08-00530-f001:**
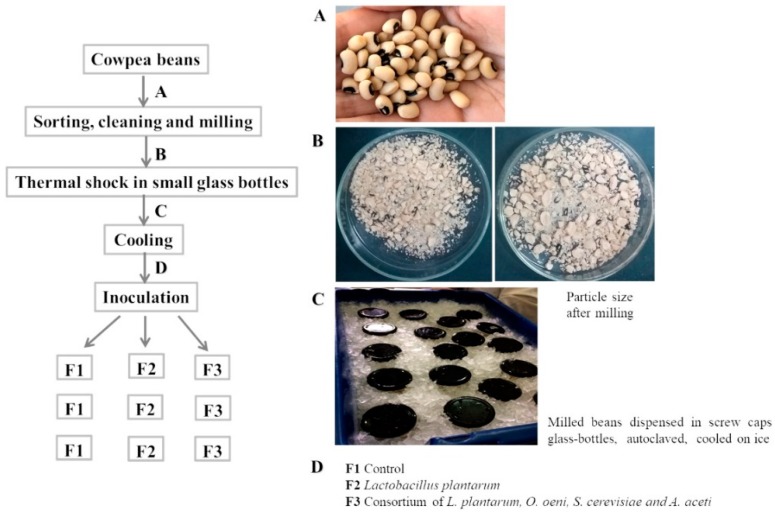
Flowchart showing the experimental design with the major steps of the fermented flour-bean production.

**Table 1 foods-08-00530-t001:** Comparative analysis of the nutritional composition (g·kg^−1^ DM) of cowpea beans and other similar legumes.

Constituent	Cowpea Beans	Fenugreek Seeds	Cowpea beans *oraludi*	Whole Cowpea Flour	Unfermented Cowpea Beans Flour Orituco	Unfermented Cowpea Beans Flour Tuy	Chickpea	Cowpea Beans	Cowpea Beans	Whole Unfermented Cowpea Beans Flour
Dry matter										957 ± 3.9
Moisture	79 ± 9.6	94 ± 11.0	78.2 ± 1.4				82 ± 4.0	74 ± 2.0	68 ± 2.0	43 ± 3.9
OM										963 ± 2.9
Carbohydrates	748 ± 28.1		493 ± 5.7	601 ± 0.6 *			611± 18	633 ± 12.0	666 ± 21.0	
Total Ash	39.3 ± 1.4	34 ± 8.0	38. ± 0.8	38 ± 0.6			22 ± 0.0	29 ± 0.0	27 ± 0.0	37 ± 2.9
Crude Protein	261 ± 2.7	254 ± 12.0	262 ± 0.23	223 ± 2.0	267 ± 5.6	261 ± 1.6	237 ± 11.0	241 ± 9.0	225 ± 10.0	254 ± 8.3
Crude Fat		53 ± 8.0	80 ± 4.1	21 ± 1.0	23 ± 0.2	22 ± 0.9	48 ± 1.0	23 ± 0.0	14 ± 0.0	18 ± 6.1
TDF	26 ± 2.0	78 ± 11.0	49 ± 0.8		149 ± 0.1	194 ± 0.1	148 ± 4.0	141 ± 3.0	163 ± 5.0	134 ± 31.7 **
Starch					243 ± 1.5	297 ± 6.0				387 ± 7.9
WSS										31 ± 2.1
References	Rivas-Vega et al. [9]	Ali et al. [30]	Ayougu et al. [10]	Khalid et al. [11]	Granito et al. [17]	Granito et al. [17]	Sreerama et al. [31]	Sreerama et al. [31]	Sreerama et al. [31]	In this study (*n* = 18)

* Carbohydrates estimate by difference [11]; WSS—water-soluble sugars; ** Crude fiber was estimated by chemical-gravimetric method, except in Granito et al., [16] and in this study where Total dietary fibre (TDF), including soluble and insoluble fractions, were determined using an enzymatic assay procedure (K-TDFR 04/17, Megazyme, Bray, Ireland) following the procedures of AOAC (2006, nº 991.43), as described in Material and methods.

**Table 2 foods-08-00530-t002:** Evolution of pH and chemical composition (g·kg^−1^ DM) of cowpea beans-flour not inoculated (F1, control), subjected to controlled fermentation with a single culture of *Lactobacillus plantarum* (F2) or inoculated with a consortium of *L. plantarum*, *Oenococcus oeni, Saccharomyces cerevisiae* and *Acetobacter aceti* (F3) along different times of fermentation (24, 48, 72 h). All experiments presented as mean values and respective standard deviation (*n* = 18).

Fermentation	Time	pH	Moist	Ash	Protein	Fat	TDF	Starch	WSS
**F1**	24	6.06 ± 0.13	42 ± 3.9	36 ± 0.0	256 ± 10.3	25 ± 11.7	164 ± 1.3	390 ± 7.8	32 ± 0.0
	48	6.04 ± 0.00	45 ± 2.8	37 ± 1.1	257 ± 1.4	27 ± 2.7	173 ± 4.3	390 ± 1.4	31 ± 3.3
	72	5.97 ± 0.31	42 ± 2.8	41 ± 7.0	258 ± 4.5	20 ± 2.9	170 ± 5.4	374 ± 3.0	31 ± 1.0
**F2**	24	5.17 ± 0.47	49 ± 2.8	36 ± 0.3	257 ± 2.1	16 ± 3.0	106 ± 6.7	389 ± 7.7	31 ± 1.7
	48	5.57 ± 0.13	46 ± 0.2	36 ± 0.1	253 ± 11.6	17 ± 3.8	108 ± 2.3	377 ± 7.1	28 ± 2.6
	72	5.26 ± 0.16	40 ± 2.1	36 ± 0.5	253 ± 9.6	15 ± 0.9	106 ± 1.1	395 ± 4.8	28 ± 5.6
**F3**	24	5.38 ± 0.14	41 ± 3.5	37 ± 0.8	238 ± 10.7	14 ± 0.8	106 ± 1.9	386 ± 6.2	33 ± 4.4
	48	5.29 ± 0.11	42 ± 5.5	33 ± 5.1	258 ± 2.2	13 ± 1.5	130 ± 1.1	394 ± 7.4	29 ± 6.5
	72	5.02 ± 0.33	38 ± 0.8	37 ± 0.6	258 ± 6.3	12 ± 1.1	146 ± 4.94	390 ± 1.7	33 ± 10.7

Experiment 0: cowpea-beans flours fermented under conditions of 20% of moisture. Moist: moisture; TFD: Total dietary fiber; WSS: Water-soluble sugars.

**Table 3 foods-08-00530-t003:** Evolution of pH, chemical composition (g·kg^−1^ DM) and nutritional characteristics of cowpea beans-flour not inoculated (F1, control), subjected to controlled fermentation with a single culture of *Lactobacillus plantarum* (F2) or inoculated with a consortium of *L. plantarum*, *Oenococcus oeni, Saccharomyces cerevisiae* and *Acetobacter aceti* (F3) along different times of fermentation (24, 48, 96 h). All experiments presented as mean values and respective standard deviation (*n* = 18).

Fermentation	Time	pH	TA	Moist	Ash	Protein	TDF	Starch	WSS
**F1**	24	5.97 ± 0.03	0.19 ± 0.01	297.3 ± 1.9	37.5 ±2.5	230.9 ± 24.9	178.2 ± 21.8	383.7 ± 10.7	60.3 ± 0.8
	48	5.98 ± 0.00	0.25 ± 0.13	323.5 ± 35.6	40.1 ± 1.6	244.1 ± 9.0	190.3 ± 25.4	374.0 ± 30.8	52.2 ± 2.0
	96	6.01 ± 0.06	0.24 ± 0.04	291.8 ± 18.0	38.7 ±1.4	238.1 ± 6.9	175.3 ± 7.2	396.5 ± 5.0	48.6 ± 0.4
**F2**	24	5.78 ± 0.08	0.17 ± 0.04	314.9 ± 18.8	38.6 ±6.1	244.5 ± 1.9	182.5 ±1.9	386.5 ±5.7	62.1 ± 5.5
	48	5.25 ± 0.09	0.25 ± 0.04	308.8 ± 34.7	36.0 ±1.7	247.2 ± 1.0	175.6 ± 35.7	343.6 ± 44.9	54.6 ± 12.2
	96	4.65 ± 0.08	0.58 ± 0.13	336.2 ± 30.7	38.9 ± 0.6	260.0 ± 0.9	207.2 ± 2.2	342.7 ± 20.9	36.7 ± 10.9
**F3**	24	5.92 ± 0.11	0.21 ± 0.03	316.1 ± 11.5	35.3 ± 2.4	245.4 ± 0.3	192.3 ± 2.1	393.7 ± 6.0	52.6 ± 7.0
	48	5.74 ± 0.01	0.21 ± 0.03	312.7 ± 10.5	38.5 ±2.5	249.4 ± 0.6	197.4 ± 3.9	393.4 ± 26.9	44.6 ± 0.6
	96	5.32 ± 0.05	0.39 ± 0.13	326.1 ± 4.2	39.3 ± 1.7	256.4 ±1.8	196.2 ± 17.6	413.4 ± 0.7	30.4 ± 3.8

Experiment 1: cowpea-beans flours fermented under conditions of 30% of moisture. TA: titrable acidity, expressed in g·L^−1^ of lactic acid; Moist: moisture; TFD: Total dietary fiber; WSS: Water-soluble sugars. The evolution of microorganisms during fermentation processes shows a gradual increase of the number of CFUs grown on MRS agar in F2 assay, ranging from 10^6^ to 1.9 × 10^8^ CFUs·g^−1^ while in F3 assay that increase was not so noticeable (10^6^–10^7^ CFUs·g^−1^). The number of CFUs grown on DRBC agar maintained constant along the fermentation process in F3, about 9 × 10^5^ CFUs·g^−1^. However, at 30% of humidity, epiphytic microorganisms grew in a non-inoculated assay (F1) after 96 h of incubation and typical colonies of lactic acid bacteria were observed on MRS agar. These results suggest that for this level of humidity, the thermal shock used (65 °C, 10 min) was insufficient to prevent the growth of natural microbiota. Even though, the number of CFUs in this case (6 × 10^2^ CFUs·g^−1^) was much lower than the one obtained in F2 (10^8^ CFUs·g^−1^).

**Table 4 foods-08-00530-t004:** Evolution of pH, TA, and nutritional characteristics (g·kg^−1^ DM) of cowpea beans-flour not inoculated (F1, control), subjected to controlled fermentation with a single culture of *Lactobacillus plantarum* (F2) or inoculated with a consortium of *L. plantarum*, *Oenococcus oeni, Saccharomyces cerevisiae* and *Acetobacter aceti* (F3) along different times of fermentation (24, 48, 96 h). All experiments presented as mean values and respective standard deviation (*n* = 18).

Fermentation	Time	pH	TA	Moist	Ash	Protein	TDF	Starch	WSS	LA
**F1**	24	6.66 ± 0.01	0.37 ± 0.06	283.3 ± 1.3	38.7 ± 0.5	244.9 ± 6.6	244.9 ± 6.6	443.2 ± 59.0	38.7 ± 0.8	0.00
	48	6.64 ± 0.02	0.37 ± 0.01	289.7 ± 3.0	39.4 ± 0.8	218.8 ± 10.4	218.8 ± 10.4	501.9 ± 5.4	32.4 ± 3.3	20.00
	96	6.46 ± 0.02	0.51 ± 0.01	290.0 ± 6.7	39.3 ± 1.8	246.2 ± 2.7	246.2 ± 2.7	522.3 ± 23.1	26.4 ± 2.3	80.00
**F2**	24	5.73 ± 0.28	0.61 ± 0.06	281.0 ± 6.1	37.2 ± 2.1	246.0 ± 5.6	246.0 ± 5.6	515.1 ± 9.6	36.7 ± 0.7	135.00
	48	4.88 ± 0.01	0.91 ± 0.01	290.6 ± 3.0	36.9 ± 1.2	244.0 ± 0.2	244.0 ± 0.2	499.4 ± 6.12	28.0 ± 0.8	390.00
	96	4.54 ± 0.04	1.20 ± 0.04	295.7 ± 3.3	38.1 ± 2.8	220.3 ± 13.5	220.3 ± 13.5	411.4 ± 17.0	22.0 ± 2.5	610.00
**F3**	24	6.11 ± 0.16	0.47 ± 0.02	289.4 ± 11.3	40.0 ± 0.7	244.5 ± 1.1	244.5 ± 1.1	372.4 ± 2.3	28.1 ± 4.0	15.00
	48	5.11 ± 0.28	0.87 ± 0.06	263.8 ± 26.3	39.1 ± 0.4	213.1 ± 13.1	213.1 ± 13.1	393.5 ± 12.4	21.1 ± 3.7	210.00
	96	4.76 ± 0.01	1.28 ± 0.06	292.2 ± 12.5	39.8 ± 0.5	235.4 ± 4.0	235.4 ± 4.0	395.6 ± 6.9	13.5 ± 3.3	410.00

Experiment 2: cowpea-beans flours fermented under conditions of 30% of moisture.TA: titrable acidity, expressed in g·L^−1^ of lactic acid; Moist: moisture; TFD: Total dietary fiber; WSS: Water-soluble sugars; LA: lactic acid expressed in mg/L.

**Table 5 foods-08-00530-t005:** Effect of time, experiment and fermentation type on chemical composition, nutritional characteristics, total phenolic content (TPC), total flavonoid content (TFC) and antioxidant activity of three hydro-alcoholic flour extracts, based on the ability to reduce ferric iron (Fe^3+^) to ferrous iron (Fe^2+^) (FRAP) and α-diphenyl-β-picrylhydrazyl (DPPH^•^) free radical scavenging method of cowpea beans-flour.

	*P*-Values			
	Time (T) *	T × Fermentation (F) §	T × Experiment (E) *	T × F × E §
**pH**	<0.001	0.0008	0.0098	0.0115
**TA**	<0.001	0.0083	0.0031	0.0115
**Moist**	0.5504	0.4748	0.8722	0.7648
**Ash**	0.4104	0.2871	0.7833	0.5417
**Protein**	0.0268	0.0690	0.0078	0.0126
**TDF**	0.8175	0.3832	0.2798	0.1180
**Starch**	0.9063	0.0074	0.3663	0.0915
**WSS**	<0.001	0.0062	0.0027	0.0114
**TPC**	0.0010	0.0366	0.0723	0.0495
**TFC**	<0.001	<0.001	0.0028	0.0133
**FRAP**	0.0075	0.0363	0.0479	0.5112
**DPPH**	<0.001	0.0145	0.6396	0.0050

* *F* test; § Wilks’ Lambda test; TA: titrable acidity, expressed in g·L^−1^ of lactic acid; Moist: moisture; TFD: Total dietary fiber; WSS: Water-soluble sugars; LA: lactic acid expressed in mg/L; TPC—expressed as µg of gallic acid equivalents (GAE)/100 mg of flour dry matter (DM); TFC—µg of catechin equivalents (CAE)/100 mg of flour DM; FRAP—µmol ferrous sulphate equivalents (FSE)/100 mg of flour DM; DPPH^•^—µg of trolox equivalents (TE)/100 mg of flour DM.

**Table 6 foods-08-00530-t006:** Total phenolic content (TPC), total flavonoid content (TFC) and antioxidant activity of three hydro-alcoholic flour extracts, based on the ability to reduce ferric iron (Fe^3+^) to ferrous iron (Fe^2+^) (FRAP) and α-diphenyl-β-picrylhydrazyl (DPPH^•^) free radical scavenging method. Experiments: F1, control; F2, fermentation with a single culture of *Lactobacillus plantarum* and F3–fermentation with a consortium of *L. plantarum*, *Oenococcus oeni, Saccharomyces cerevisiae* and *Acetobacter aceti* along different times of fermentation (24, 48, 96 h). All experiments presented as mean values and respective standard deviation (*n* = 18).

Fermentation	Time	TPC	TFC	FRAP	DPPH^•^
**F1**	24	60.32 ± 20.17	39.22 ± 4.14	0.87 ± 0.14	66.17 ± 6.41
	48	130.97 ± 8.45	65.74 ± 4.55	1.27 ± 0.08	118.65 ± 10.30
	96	67.20 ± 3.49	32.07 ± 3.41	1.03 ± 0.10	49.72 ± 6.51
**F2**	24	93.89 ± 9.18	36.81 ± 3.16	1.25 ± 0.10	88.50 ± 7.89
	48	117.71 ± 6.08	53.31 ± 3.64	1.50 ± 0.11	109.26 ± 9.09
	96	119.57 ± 8.76	52.63 ± 6.38	1.45 ± 0.10	91.85 ± 5.16
**F3**	24	69.22 ± 14.36	31.80 ± 3.12	0.97 ± 0.16	48.91 ± 5.08
	48	60.42 ± 6.74	30.37 ± 2.93	0.83 ± 0.05	50.36 ± 6.31
	96	63.56 ± 19.61	23.35 ± 2.82	0.56 ± 0.07	64.26 ± 27.06

Experiment 1: cowpea-beans flours fermented under conditions of 30% of moisture.TPC—expressed as µg of gallic acid equivalents (GAE)/100 mg of flour dry matter (DM); TFC—µg of catechin equivalents (CAE)/100 mg of flour DM; FRAP—µmol ferrous sulphate equivalents (FSE)/100 mg of flour DM; DPPH^•^ -µg of trolox equivalents (TE)/100 mg of flour DM.

**Table 7 foods-08-00530-t007:** Total phenolic content (TPC), total flavonoid content (TFC) and antioxidant activity of three hydro-alcoholic flour extracts, based on the ability to reduce ferric iron (Fe^3+^) to ferrous iron (Fe^2+^) (FRAP) and α-diphenyl-β-picrylhydrazyl (DPPH^•^) free radical scavenging method. Experiments: F1, control; F2, fermentation with a single culture of *Lactobacillus plantarum* and F3—fermentation with a consortium of *L. plantarum*, *Oenococcus oeni, Saccharomyces cerevisiae* and *Acetobacter aceti* along different times of fermentation (24, 48, 96 h). All experiments presented as mean values and respective standard deviation (*n* = 18).

Fermentation	Time	TPC	TFC	FRAP	DPPH^•^
**F1**	24	104.38 ± 11.32	43.65 ± 5.07	1.19 ± 0.08	62.46 ± 10.05
	48	128.98 ± 7.12	52.58 ± 5.45	1.49 ± 0.06	84.43 ± 7.42
	96	85.37 ± 10.05	34.88 ± 4.25	1.01 ± 0.17	51.80 ± 5.26
**F2**	24	74.71 ± 10.25	24.48 ± 2.17	1.01 ± 0.15	53.11 ± 4.03
	48	108.71 ± 6.63	40.73 ± 3.81	1.25 ± 0.11	61.28 ± 3.84
	96	98.76 ± 25.93	31.37 ± 5.92	1.04 ± 0.14	53.79 ± 4.48
**F3**	24	111.56 ± 25.93	46.42 ± 5.52	1.18 ± 0.06	95.62 ± 8.64
	48	149.12 ± 6.20	38.37 ± 2.77	1.19 ± 0.10	138.30 ± 11.85
	96	86.93 ± 7.71	25.86 ± 2.69	0.53 ± 0.04	89.86 ± 6.13

Experiment 2: cowpea-beans flours fermented under conditions of 30% of moisture. TPC—expressed as µg of gallic acid equivalents (GAE)/100 mg of flour dry matter (DM); TFC—µg of catechin equivalents (CAE)/100 mg of flour DM; FRAP—µmol ferrous sulphate equivalents (FSE)/100 mg of flour DM; DPPH^•^—µg of trolox equivalents (TE)/100 mg of flour DM.

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
