# Peer review of "Effect of Controlled Microbial Fermentation on Nutritional and Functional Characteristics of Cowpea Bean Flours"

_foods, 2019, doi:10.3390/foods8110530_

Round 1

Reviewer 1 Report

Referee's Comments:

The article entitled "Effect of controlled microbial fermentation on nutritional and functional characteristics of cowpea bean flours" is about the study of different fermentations of cowpea bean flour, using different microorganisms and different temperature and humidity conditions, to obtain a fermented product with a greater number of assimilable compounds, safer from the microbiological point of view and with new sensory attributes.

Suggestions:

In this sense, the following are suggestions to be taken into account to improve the quality of the text of the manuscript.

Abstract:

- Line 26: Write Saccharomyces cerevisiae in italics.
- Lines 30, 31, 32: Write the P in italics or substitute p o ρ so as not to confuse the degree of significance with the P value corresponding to the statistical analysis of the data, as reflected in line 35.
- Line 32: Write the full name L. plantarum, as this is the first time it has been named in the manuscript and in italics.

Introduction:

- Line 46: Write Vigna unguiculata in italics.
- Line 53: From now on, it is suggested to use superscript in the units. It is clearer to the reader and will facilitate the reading of the text. Example: g.Kg-1
- Line 53: There is a double space in front of the appointments [10, 11]
- Line 55: There is a double space in front of the appointments [1, 12]
- Line 56: There is a double space in front of the appointments [13]
- Lines 58-60: Facts of importance are affirmed that should be cited results published in bibliography.
- Lines 60-61: Facts of importance are affirmed that should be cited results published in bibliography.
- Lines 63-64: Facts of importance are affirmed that should be cited results published in bibliography.
- Line 69: Insert the quotation in the text (Granito et al., 2005) with the quotation style used in Foods.
- Lines 71-72: Affirm facts of importance that should be cited from results published in bibliography.

2. Materials and methods:

- Justify text in all subsections.

2.2. Preparation of cowpea bean flours:

- Lines 89-92: Has the preparation of the different flour samples been carried out under controlled conditions of temperature and lighting? It is not clear in the text and may be important factors in the handling of flours, where some components such as fatty acids are sensitive to oxidation processes. This could affect the nutritional and sensory characteristics of the samples.
- Lines 92-94: It would be interesting to include in the text the specifications of the autoclave used, given that in some cases it has been worked under pasteurization conditions.
- Line 99: There is a double space after 72h.

2.3. Microorganisms:

- Lines 111, 112: Write in subscript 2 of CO2. Review the rest of the manuscript because there is the same error in different subsections.
- Line 115: Correct in the text the symbol used to indicate the degrees Celsius. Check the rest of the manuscript and correct every time it is written incorrectly.
- Lines 118-119: Write Saccharomyces cerevisiae in italics.

2.5. Microbiological analysis:

- Line 143: Citations are marked in bold in the text. Review the rest of the manuscript because the same error exists in different subsections.

2.8. Statistical analysis:

- This sub-section should include the level of significance of the test that has been applied in the statistical study.
- Have the authors checked that the data studied conform to a normal distribution before applying a MANOVA analysis? In this case, if the results have been standardised prior to their statistical analysis, it would be useful to indicate which method they followed for the verification and the result obtained.

4. CONCLUSIONS:

- The conclusions of the manuscript are very general, it does not give more concrete results obtained by the researchers. This makes it lose impact and interest for the reader.

Tables (1-6):

- It should reflect the significant differences found in the results when comparing the different samples or experiments carried out. In this sense, this would greatly enrich the scientific quality of the work done. Furthermore, in the case of such inclusion, the statistical treatment applied and its interpretation should be clearly indicated at the foot of the tables.

Figure 1:

- Figure 1 is of very poor quality in design and in the information it provides to the manuscript. It could be eliminated from the work and even so, relevant information would not be eliminated from it.

Bibliographical references:

- First, check that the bibliography is well aligned in the text.
- Secondly, more current bibliography should be included to give more interest to the work. In this sense, authors could take advantage of the recommendation to include citations in previous subsections.

Comments to Author:

- For future work, it would be very interesting to carry out a descriptive sensory analysis of the wine samples. The information provided by the sensory analysis could make it possible to demonstrate organoleptic differences between the different samples under study. In this sense, the work could be much more complete with this information from enological point of view.
- In addition, a study of the useful life of already processed flours could be included to corroborate some of the conclusions that they assert.

Author Response

The authors  want to thank the reviewers for their time and their positive comments on our article, as well as the notes performed which allowed us to improve the quality of the manuscript. Below are detailed point-by-point answers to the issues raised by the reviewer 1

The article entitled "Effect of controlled microbial fermentation on nutritional and functional characteristics of cowpea bean flours" is about the study of different fermentations of cowpea bean flour, using different microorganisms and different temperature and humidity conditions, to obtain a fermented product with a greater number of assimilable compounds, safer from the microbiological point of view and with new sensory attributes.

Suggestions: In this sense, the following are suggestions to be taken into account to improve the quality of the text of the manuscript.

Abstract:

- Line 26: Write Saccharomyces cerevisiae in italics.

Authors : Done. All names of microbial species are now written in italics all over the text

- Lines 30, 31, 32: Write the P in italics or substitute p o ρ so as not to confuse the degree of significance with the P value corresponding to the statistical analysis of the data, as reflected in line 35.

Authors : Done. P is written in italics as suggested all over the text

- Line 32: Write the full name L. plantarum, as this is the first time it has been named in the manuscript and in italics.

Authors : Done. All names of microbial species are now written in italics all over the text

Introduction:

- Line 46: Write Vigna unguiculata in italics.

Authors : Done. All names of plant species are now written in italics all over the text

- Line 53: From now on, it is suggested to use superscript in the units. It is clearer to the reader and will facilitate the reading of the text. Example: g.Kg-1

Authors : Superscript in unities is used now all over the text

- Line 53: There is a double space in front of the appointments [10, 11]

Authors : Done

- Line 55: There is a double space in front of the appointments [1, 12]

Authors : Done

-Line 56: There is a double space in front of the appointments [13]

Authors : Done

- Lines 58-60: Facts of importance are affirmed that should be cited results published in bibliography.

Authors: Reference number [2] included FAO 2004. COWPEA: Post-Harvest Operations. Food and Agriculture Organization of the United Nations (FAO), Rome, Italy. Carlos Gómez Author: Edited by AGST/FAO: Danilo Mejía, PhD, FAO (Technical). http://www.fao.org/3/a-au994e.pdf.

- Lines 60-61: Facts of importance are affirmed that should be cited results published in bibliography.- Lines 63-64: Facts of importance are affirmed that should be cited results published in bibliography.; - Lines 71-72: Affirm facts of importance that should be cited from results published in bibliography.

Authors: References have been included in the text:

Marco ML., Heeney D., Binda S., ….. Anne Pihlanto A., Smid E.J.,  Hutkins R. 2017. Health benefits of fermented foods: microbiota and beyond.  Current Opinion in Biotechnology 44, 94–102. [14]

Blackburn C. de W. 2006. Food Spoilage Microorganisms. Woodhead Publishing Series in Food Science, Technology and Nutrition, pp xvii-xxiii. https://doi.org/10.1016/B978-1-85573-966-6.50027-4. [20]

Rawat S. 2015. Food Spoilage: Microorganisms and their prevention. Asian J Plant Sci. Res. 5, 47-56 [21]

- Line 69: Insert the quotation in the text (Granito et al., 2005) with the quotation style used in Foods.

Authors : The citation was replaced by the quotation style used in Foods - [16]

Materials and methods:

- Justify text in all subsections.

Authors : Text is now justified in all section of M&M

2.2. Preparation of cowpea bean flours:

- Lines 89-92: Has the preparation of the different flour samples been carried out under controlled conditions of temperature and lighting? It is not clear in the text and may be important factors in the handling of flours, where some components such as fatty acids are sensitive to oxidation processes. This could affect the nutritional and sensory characteristics of the samples.

Authors: We understand the question raised by the reviewer. Indeed milling procedure was done under well controlled laboratory conditions: no temperature variation (20°C) nor exposure to direct lighting happened during the process.

- Lines 92-94: It would be interesting to include in the text the specifications of the autoclave used, given that in some cases it has been worked under pasteurization conditions. 

Authors : The characteristics of the autoclave are now included in line 94: 60L-stainless steel steam autoclave, Orion B6, GA, USA)

- Line 99: There is a double space after 72h.

Authors : Removed the double space

2.3. Microorganisms:

- Lines 111, 112: Write in subscript 2 of CO2. Review the rest of the manuscript because there is the same error in different subsections.

Authors : Corrected all over the text

- Line 115: Correct in the text the symbol used to indicate the degrees Celsius. Check the rest of the manuscript and correct every time it is written incorrectly.

Authors : Corrected all over the text

- Lines 118-119: Write Saccharomyces cerevisiae in italics.

Authors : Done. All names of microbial species are now written in italics all over the text

2.5. Microbiological analysis:

- Line 143: Citations are marked in bold in the text. Review the rest of the manuscript because the same error exists in different subsections.

Authors : Corrected in all sections where the same mistake existed.

2.8. Statistical analysis:

- This sub-section should include the level of significance of the test that has been applied in the statistical study.

Authors : The level of significance of the test was included in the text (lines 229-230).

- Have the authors checked that the data studied conform to a normal distribution before applying a MANOVA analysis? In this case, if the results have been standardised prior to their statistical analysis, it would be useful to indicate which method they followed for the verification and the result obtained.

Authors : Yes, before the analysis normality of the data was tested using the Shapiro-Wilk test (statement now included in the text lines 230-231). Only TA, Protein and Starch variables did not follow the normal distribution and, consequently, they were transformed (log10 function was used for data transformation) before the analysis. Statement now included in the text (lines 370-371)

CONCLUSIONS:

- The conclusions of the manuscript are very general, it does not give more concrete results obtained by the researchers. This makes it lose impact and interest for the reader.

Authors : Conclusions have been rewritten to highlight major results obtained in this study: The inoculation of cowpea bean flours with a single culture or a consortium of microorganisms revealed how fermentation influenced the physicochemical and possibly the functional properties of this new product. In fact, in this study, pH, total acidity, dry matter, TDF, WSC and protein were found to be altered in cowpea bean flours. Also, phytochemicals content, namely phenolic compounds and flavonoids, and the results for antioxidant activity assays suffered changes with microbial fermentation. The high content of dietary fibre of this product can have numerous therapeutic and beneficial effects in the organism due to stimulation of the gastrointestinal microbiota, functioning as a prebiotic and enhancing human health. Moreover, it is a plant-based protein source which meets current trends in human diets with increasing interest in natural products incorporation and following flexitarianism, vegetarianism or veganism food patterns. It may therefore be nutritionally interesting to ferment cowpea beans before their use as a functional ingredient, since the fermentation process produced a more stable and safer product with potential to be incorporated in new foodstuffs.

Tables (1-6):

- It should reflect the significant differences found in the results when comparing the different samples or experiments carried out. In this sense, this would greatly enrich the scientific quality of the work done. Furthermore, in the case of such inclusion, the statistical treatment applied and its interpretation should be clearly indicated at the foot of the tables.

Authors : As suggested by the reviewer data on the statistical analysis, and respective footnotes, was included in a new table (Tables 7). However, as Table 1 in only a comparative analysis of the nutritional composition of cowpea beans with other similar legumes and Table 2 shows data on the effect of using 20% of humidity on the microbial fermentation in bean flours (these results can be seen as a preliminary experiment as we observed a lack of microbial activity under these conditions) no modification was performed on these two tables.

Figure 1:

- Figure 1 is of very poor quality in design and in the information it provides to the manuscript. It could be eliminated from the work and even so, relevant information would not be eliminated from it.

Authors : Figure 1 was replaced by one with a better design and with more information about the experimental work

Bibliographical references:

- First, check that the bibliography is well aligned in the text.

Authors : Done

- Secondly, more current bibliography should be included to give more interest to the work. In this sense, authors could take advantage of the recommendation to include citations in previous subsections.

Authors: References have been included in the text: Marco ML., Heeney D., Binda S., ….. Anne Pihlanto A., Smid E.J.,  Hutkins R. 2017. Health benefits of fermented foods: microbiota and beyond.  Current Opinion in Biotechnology 44, 94–102. [14]: Backburn C. de W. 2006. Food Spoilage microorganisms  Woodhead Publishing Series in Food Science, Technology and Nutrition , pp xvii-xxiii. https://doi.org/10.1016/B978-1-85573-966-6.50027-4. [20];Rawat S. 2015. Food Spoilage: Microorganisms and their prevention. Asian J Plant Sci. Res. 5, 47-56 [21]

Comments to Author:

- For future work, it would be very interesting to carry out a descriptive sensory analysis of the wine samples. The information provided by the sensory analysis could make it possible to demonstrate organoleptic differences between the different samples under study. In this sense, the work could be much more complete with this information from enological point of view.

Authors : Authors are aware of the importance of sensory analysis as an important attribute to define the quality of a specific food product. In some food products or beverages some differences are better detected by sensory analysis rather than by chemical/chromatographic analysis. Considering the promising results obtained herein further research on the subject will be conducted, then the impact of microbial fermentation on chemical composition and sensory attributes will evaluated

- In addition, a study of the useful life of already processed flours could be included to corroborate some of the conclusions that they assert.

Authors : the authors appreciate the suggestion and we will have into consideration in further works 

Reviewer 2 Report

The work “Effect of controlled microbial fermentation on nutritional and functional characteristics of cowpea bean flours” is interesting and the experimental plan is clearly described. Since several years the taxonomy of the genus Leuconostoc has been changed and in particular Leuconostoc oenos is today the species Oenococcus oeni. Authors must change it in the text. Each species mentioned must be in italics. I have corrected this in several places, but the authors must review in detail.

The methods are also well described. The discussion is well argued in comparison to the results. I would recommend the authors to implement the part concerning the results of this work because sometimes there is confusion with those reported in the bibliography.

Particular improvement is necessary for the conclusions: too narrow. In general this work is innovative and has interesting potential applications.

Minor comments:

Line 29 …… WSC add in the brackets (total water-soluble carbohydrate (WSC)

Line 108 Coirrect in……. cultures two strains of lactic acid bacteria (LAB) species, Lactobacillus plantarum (Lp),

Lines 111, 112 CO2   Correct in CO2

Line 112 Correct in an acetic acid bacteria …

Line 118 Bac??’ Correct in Aac

Lines 118-119 S. cerevisiae in italic

Line 124 …concentration of 106 mL–1. C   Correct in 106 mL–1

Lines 130-133 Correct in: ……… in each assay; F3 (Sc-Lp-Lo-Aac): In this formulation, a consortium of the same strain of LAB used in F2 experiment, L.plantarum (Lp) plus O.oeni (Lo), S. cerevisiae (Sc) and A. aceti (Aac) was used as inoculum. Each one was separately grown in appropriate culture media, MRS, YPD or BHI broth

Line 154… units (CFU) g−1 sample Correct in units (CFU) g−1 sample

Line 164 …with 800 mL.L−1 ethanol.. Correct in with 800 mL.L−1 ethanol

Line 174 Eliminate the brackert …10 g of (fermented flours

Line 191 Na2CO3 Correct in Na2CO3

Line 194 Gallic acid (5-100 mg.L−1)   Correct in Gallic acid (5-100 mg.L−1)

Line 201 NaNO2 ..Correct in NaNO2   AlCl3   in AlCl3

Line 202.. 2 mL of 1 mol.L−1 NaOH   Correct in 2 mL of 1 mol.L−1 NaOH

Line 211 ….6.0 x 10−5 mol.L−1 in ethanol) Correct in 6.0 x 10−5 mol.L−1 in ethanol)

Line 222 ….20 mM FeCl3.6H2O solution) Correct in 20 mM FeCl3.6H2O solution)

Line 224 (FeSO4.7H2O   Correct in (FeSO4.7H2O)

Table 1 Add in the legend WSS = water soluble sugars

Line 235 …other type of legume Correct in   other types of legume

Line 238 …8.3 g.kg-1 DM Correct in 8.3 g.kg-1 DM

Line 240 Vigna unguiculata in corsivo

Line 244 …..17.7 ± 6.1 g.kg-1 Correct in 17.7 ± 6.1 g.kg-1

Line 246 79.8 g.kg−1 .. Correct in 79.8 g.kg−1 DM

Line 246 Vigna unguiculata …in italics

Line 247 Vigna biflorus and Vigna sinensis in italics

Line 248 ….418 g.kg-1…Correct in 418 g.kg-1

Line 249 …493 g.kg-1….. Correct in 493 g.kg-1 DM

Line 252 …297 g.kg−1 …..Correct in 297 g.kg−1 DM

Line 254….133.6 ± 31.7 g.kg-1……Correct in 133.6 ± 31.7 g.kg-1

Line 255….194 g.kg−1 ……Correct in 194 g.kg−1

Line 268 ……chemical and microbiologically   Correct in chemically and microbiologically

Line 269 …….result Correct in results

Line 293 These differences can be due to several factors: also by the strains used as starter for the fermentation. This aspect is important to underline in the text. It is widely demonstrated different results in process parameters in function of the different strains used.

Line 296 Correct in: The evolution of microorganisms during fermentation processes shows

Line 297… 106 to 1.9 x 108 CFUs.g-1 Correct in 106 to 1.9 x 108 CFUs.g-1

Line 298… (106 to 107 CFUs.g-1)….Correct in (106 to 107 CFUs.g-1)

Line 298 RBCA ?

Line 299 ….9x105 CFUs.g-1 Correct in 9x105 CFUs.g-1

Lines 303-304 …6x102 CFUs.g-1) Correct in 6x102 CFUs.g-1)  

       (108 CFUs.g-1 Correct in . (108 CFUs.g-1)

Lines 305-307 This sentence is confused. Reformulate the concept.

Line 315    Vigna sinensis   in italics

Line 360 …Fe3+/ferricyanide …. . (Fe2+) Correct in Fe3+/ferricyanide ……(Fe2+),

Line 370 ….scavenging assay was observed…Correct in… scavenging assay were observed

Line 373 …reported by (Zhao et al., 2014) …Correct in reported (34).

Author Response

The authors  want to thank the reviewers for their time and their positive comments on our article, as well as the notes performed which allowed us to improve the quality of the manuscript. Below are detailed point-by-point answers to the issues raised by the reviewer 2

The work “Effect of controlled microbial fermentation on nutritional and functional characteristics of cowpea bean flours” is interesting and the experimental plan is clearly described. Since several years the taxonomy of the genus Leuconostoc has been changed and in particular Leuconostoc oenos is today the species Oenococcus oeni. Authors must change it in the text. Each species mentioned must be in italics. I have corrected this in several places, but the authors must review in detail.

The methods are also well described. The discussion is well argued in comparison to the results. I would recommend the authors to implement the part concerning the results of this work because sometimes there is confusion with those reported in the bibliography.

Particular improvement is necessary for the conclusions: too narrow. In general this work is innovative and has interesting potential applications.

Minor comments:

Line 29 …… WSC add in the brackets (total water-soluble carbohydrate (WSC)

Authors: done

Line 108 Correct in……. cultures two strains of lactic acid bacteria (LAB) species, Lactobacillus plantarum (Lp),

Authors: the word strains was included

Lines 111, 112 CO2   Correct in CO2

Authors: corrected all over the text

Line 112 Correct in an acetic acid bacteria …

Authors: done

Line 118 Bac??’ Correct in Aac

Authors: Bac was changed to Aac, now in line 120.

Lines 118-119 S. cerevisiae in italic

Authors: done

Line 124 …concentration of 106 mL–1. C   Correct in 106 mL–1

Authors: corrected all over the text

Lines 130-133 Correct in: ……… in each assay; F3 (Sc-Lp-Lo-Aac): In this formulation, a consortium of the same strain of LAB used in F2 experiment, L.plantarum (Lp) plus O.oeni (Lo), S. cerevisiae (Sc) and A. aceti (Aac) was used as inoculum. Each one was separately grown in appropriate culture media, MRS, YPD or BHI broth

Authors: The setence was corrected according to the suggestion

Line 154… units (CFU) g−1 sample Correct in units (CFU) g−1 sample

Authors: Superscript was inserted

Line 164 …with 800 mL.L−1 ethanol.. Correct in with 800 mL.L−1 ethanol

Authors: Superscript was inserted

Line 174 Eliminate the brackert …10 g of (fermented flours

Authors: Done

Line 191 Na2CO3 Correct in Na2CO3

Authors: Done

Line 194 Gallic acid (5-100 mg.L−1)   Correct in Gallic acid (5-100 mg.L−1)

Authors: Done

Line 201 NaNO2 ..Correct in NaNO2   AlCl3   in AlCl3

Authors: Done

Line 202.. 2 mL of 1 mol.L−1 NaOH   Correct in 2 mL of 1 mol.L−1 NaOH

Authors: Done

Line 211 ….6.0 x 10−5 mol.L−1 in ethanol) Correct in 6.0 x 10−5 mol.L−1 in ethanol)

Authors: Done

Line 222 ….20 mM FeCl3.6H2O solution) Correct in 20 mM FeCl3.6H2O solution)

Authors: Done

Line 224 (FeSO4.7H2O   Correct in (FeSO4.7H2O)

Authors: Done

Table 1 Add in the legend WSS = water soluble sugars

Authors: Done

Line 235 …other type of legume Correct in   other types of legume

Authors: Done

Line 238 …8.3 g.kg-1 DM Correct in 8.3 g.kg-1 DM

Authors: Done

Line 240 Vigna unguiculata in corsivo

Authors: Done

Line 244 …..17.7 ± 6.1 g.kg-1 Correct in 17.7 ± 6.1 g.kg-1

Authors: Done

Line 246 79.8 g.kg−1 .. Correct in 79.8 g.kg−1 DM

Authors: Done

Line 246 Vigna unguiculata …in italics

Authors: Done

Line 247 Vigna biflorus and Vigna sinensis in italics

Authors: Done

Line 248 ….418 g.kg-1…Correct in 418 g.kg-1; Line 249 …493 g.kg-1….. Correct in 493 g.kg-1 DM; Line 252 …297 g.kg−1 …..Correct in 297 g.kg−1 DM; Line 254….133.6 ± 31.7 g.kg-1……Correct in 133.6 ± 31.7 g.kg-1; Line 255….194 g.kg−1 ……Correct in 194 g.kg−1

Authors: Superscript was inserted for all unities; The mistake was corrected all over the text

Line 268 ……chemical and microbiologically   Correct in chemically and microbiologically

Authors: Done

Line 269 …….result Correct in results

Authors: Done

Line 293 These differences can be due to several factors: also by the strains used as starter for the fermentation. This aspect is important to underline in the text. It is widely demonstrated different results in process parameters in function of the different strains used.

Authors: Authors agree and a statement was included in the text: also by the strains used as starter for the fermentation herein in this study.

Line 296 Correct in: The evolution of microorganisms during fermentation processes shows

Authors: Done

Line 297… 106 to 1.9 x 108 CFUs.g-1 Correct in 106 to 1.9 x 108 CFUs.g-1; Line 298… (106 to 107 CFUs.g-1)….Correct in (106 to 107 CFUs.g-1)

Authors: Superscript was used.

Line 298 RBCA ?

Authors: Sorry for the mistake: RBCA was replaced by DRBC the correct acronym Dichloran Rose Bengal Chloramphenicol agar

Line 299 ….9x105 CFUs.g-1 Correct in 9x105 CFUs.g-1; Lines 303-304 …6x102 CFUs.g-1) Correct in 6x102 CFUs.g-1) ;  (108 CFUs.g-1 Correct in . (10CFUs.g-1)

Authors: Superscript was used.

Lines 305-307 This sentence is confused. Reformulate the concept.

Authors: The sentence was reformulated. A new sentence was introduced in lines 444-445: It is generally accepted that microbial fermentation increases the nutritional value of legumes, by increasing the levels of essential nutrients and/or by reducing the level of antinutrients in food

Line 315    Vigna sinensis   in italics

Authors: Done

Line 360 …Fe3+/ferricyanide …. . (Fe2+) Correct in Fe3+/ferricyanide ……(Fe2+),

Authors: Done

Line 370 ….scavenging assay was observed…Correct in… scavenging assay were observed

Authors: Done

Line 373 …reported by (Zhao et al., 2014) …Correct in reported (34).

Authors : The reference was changed to the number in brackets [34].

Reviewer 3 Report

The manuscript presents some interesting results, and generally well written. A few issues, however, need to be addressed:

Material and methods section:

Line 92; the author(s) mentioned that the “ground beans were dispensed into small screw caps glass-flasks”, it will be better if there is a specific ratio

Line 108; “As fermentation-starters cultures two lactic acid bacteria (LAB) species”

It could be “As fermentation-starters cultures including; two lactic acid bacteria (LAB) species….”

Line 124; “an initial cellular concentration of 106 mL–1” is this not a high concentration for an initial count?

Line 126, and line 135; the author mentioned that “the autoclaved milled beans were watered with saline solution” this is not clear statement, is this a dilution and what is the ratio?

Line 147,148; “One ml of the suspension was serially diluted in 0.1% peptone water and 0.1 ml or 1 mL samples from each of three consecutive dilutions were spread inoculated onto duplicate plates of different agar media” this sentence is not clear enough.

Line 153; “the typical colonies of yeast or bacteria were counted” how you differentiate between strains colonies in each treatment.

In the results section:

Line 296; “ranging from 106 to 1.9 x 108 CFUs.g-1” this is a very low increase compared to the fermented time.

Figure 1 needs a little bit more improve and make sure to include all the experiment steps.

Also, it would be nice if there is a figure to present the CFU results in each treatment for the non-inoculated and inoculated assay. 

Author Response

The authors  want to thank the reviewers for their time and their positive comments on our article, as well as the notes performed which allowed us to improve the quality of the manuscript. Below are detailed point-by-point answers to the issues raised by the reviewer 3

The manuscript presents some interesting results, and generally well written. A few issues, however, need to be addressed:

Material and methods section:

Line 92; the author(s) mentioned that the “ground beans were dispensed into small screw caps glass-flasks”, it will be better if there is a specific ratio

Authors: The fermentations were conducted in small screw caps glass-flasks lled to 2/3 of their volume. This sentence appears later in line 155.

Line 108; “As fermentation-starters cultures two lactic acid bacteria (LAB) species”. It could be “As fermentation-starters cultures including; two lactic acid bacteria (LAB) species….”

Authors: The word strains was included in the paragraph as “….As fermentation-starters cultures two strains of lactic acid bacteria (LAB) species, Lactobacillus plantarum (Lp), a facultative homofermentative LAB which converts glucose almost exclusively into lactic acid, and Oenococcus oenos (Lo), a heterofermentative LAB which catabolize glucose into lactic acid, ethanol/acetate and CO2 as well as yeast strain of Saccharomyces cerevisiae (Sc), that converts the sugar into ethanol CO2, and a strain of acetic acid bacteria, Acetobacter aceti (Aac), ….

Line 124; “an initial cellular concentration of 106 mL–1” is this not a high concentration for an initial count?

Authors: For many microbial fermentation processes, the inoculum level has a substantial impact on the process performance. We have already investigated the effect of inoculum size on several parameters of alcoholic fermentation. On the basis of these observations cell concentration in the inoculum is usually kept around 106 viable cells/mL. The maximum number of cells reached at the end of the fermentation processes never exceeds 108 for wine yeast or 109  for bacteria.

Line 126, and line 135; the author mentioned that “the autoclaved milled beans were watered with saline solution” this is not clear statement, is this a dilution and what is the ratio?

Authors: The amount of sterile saline solution added to milled beans in each assay, it was calculated to obtain 10, 20 or 30% dependent on the assay as it is mentioned in lines 127-132.

Line 147,148; “One ml of the suspension was serially diluted in 0.1% peptone water and 0.1 ml or 1 mL samples from each of three consecutive dilutions were spread inoculated onto duplicate plates of different agar media” this sentence is not clear enough.

Authors: authors understand the question rised by the reviewer. In fact,  enumeration of LAB is done by spreading 1 mL on a double-layered plate of MRS agar whereas yeasts/moulds are enumerated by spreading 0.1mL of each dilution on DRBC agar . This sentence appears in lines 197-201.

Line 153; “the typical colonies of yeast or bacteria were counted” how you differentiate between strains colonies in each treatment.

Authors: Lactic acid bacteria do not growth on DRBC agar, which is used for enumeration of yeasts/moulds ISO 21527-1:2008 [23]. MRS is the culture media used for enumeration of mesophilic lactic acid bacteria for food analysis ISO 15214:1998 [22]. Typical colonies of LAB are counted in MRS.

In the results section:

Line 296; “ranging from 106 to 1.9 x 108 CFUs.g-1” this is a very low increase compared to the fermented time.

Authors: For many microbial fermentation processes, the inoculum level has a substantial impact on process performance. We have already investigated the effect of inoculum size on several parameters of alcoholic fermentation. On the basis of these observations cell concentration in the inoculum is usually kept around 106 viable cells/mL. The maximum number of cells reached at the end of the fermentation processes never exceeds 108 when yeast or 109  bacteria are used.

Figure 1 needs a little bit more improve and make sure to include all the experiment steps.

Authors : Figure 1 was replaced by one with a better design and with more information about the experimental work

Also, it would be nice if there is a figure to present the CFU results in each treatment for the non-inoculated and inoculated assay. 

Authors : The authors consider that the insertion of a new figure with these data does not provide additional information since the number of CFUs at the beginning and at the end of fermentation are presented along the text.

Round 2

Reviewer 1 Report

In general, the authors have complied with the suggestions and recommendations regarding the initial manuscript. Although, with respect to better reflecting the statistics associated with the data shown in the tables, version 2 of the manuscript still does not reflect it clearly.

In the case of Table 1 and Table 2, although these are preliminary results as indicated by the authors, it does not exempt the need to add an indicative for those treatments that present significant differences from the objective point of view of statistics. I will leave it up to the Editor to decide that the authors include the statistics in Table 1, Table 2, Table 3, Table 4, Table 6 and Table 7. In the case of Table 7, the authors say that if these indications have been added but the manuscript v2 still does not appear.